# Stability Analysis of Cavern Collapse in Fractured-Cavity Oil Reservoirs

**Yanzhi Ding [1], Qiangyong Zhang [1,*], Wen Xiang [2], Bin Wang [1], Xinrui Lyu [3] and Longyun Zhang [4]**

1   Geotechnical and Structural Engineering Research Center, Shandong University, Jinan 250100, China; 201613296@mail.sdu.edu.cn (Y.D.)
2   School of Civil Engineering, Shandong University, Jinan 250100, China
3   Petroleum Exploration and Production Research Institute, SINOPEC, Beijing 100083, China
4   Logistics Department, Shandong University, Jinan 250100, China
*   Correspondence: as.df.68@163.com

**Abstract:** Fractured-vuggy oil reservoirs, with the decrease of formation pressure during the exploitation process, lead to the collapse of caverns or the closure of sizeable fractured oil channels, which seriously affects oil well production and the recovery rate of oil reservoirs. True three-dimensional geomechanical model tests were carried out to avoid the impact of cave collapse and fracture closure on oil well production. Taking the Tahe Oilfield in Xinjiang area of China as the engineering background, we researched the collapse failure mechanism of the karstic caves in fractured-cavity oil reservoirs and the evolution of fracture closure through a true three-dimensional geomechanical model test and the numerical simulation software RFPA. The collapse failure modes of caverns with and without prefabricated cracks were revealed, along with the displacement and stress changes during cave collapse and the mechanism of cave collapse failure. Our study revealed the mechanism of the interaction between cracks and the cave. The research results show that prefabricated cracks reduce the roof of the cave's bearing capacity, making the karst cave collapse with incomplete cracks. The impact of the collapse is much smaller than the cavern without prefabricated cracks. The crack closure extends from the near end to the far end. The research results will provide necessary theoretical support for the large-scale safe extraction of deep petroleum resources, increase oil production in China, and have important theoretical significance and engineering application value.

**Keywords:** geomechanical model test; numerical simulation; caves in fractured-cavity oil reservoirs

## 1. Introduction

Carbonate reservoirs are widely distributed worldwide. According to statistics, 96 of the 236 large oil and gas fields in the world are carbonate reservoirs, accounting for about 40%, and more than 30% of carbonate reservoirs are fracture-cavity reservoirs. China's marine carbonate rock oil and gas resources are significant at $300 \times 108$ t, and petroleum resources are $150 \times 108$ t. They are mainly distributed in the Tarim Basin and North China. The fracture-cavity reservoirs account for 2/3 of the proven carbonate rock reserves, and are the main areas for increasing reserves in the future [1–5]. Carbonate fracture-cavity reservoirs are different from sandstone reservoirs. Deep burial and various types of large-scale karst caves are the main storage spaces, with different shapes and other filling characteristics. The fractures mainly serve in communication and diversion [6–8]. However, during the exploitation of the reservoir, according to the production dynamics of some oil wells, if some karst cave collapses or large, fractured oil passages occur downhole, it will seriously affect the production of oil wells. Therefore, it is of great theoretical significance and engineering application value to analyze the stability of karst caves buried deep underground. In the process of reservoir exploitation, there are mutual influences between karst caves and fractures in reservoirs. On the one hand, the collapse and destruction of

karst caves will affect fracture expansion and closure; on the other hand, fracture expansion and closure will also influence the collapse and destruction of karst caves.

At present, many studies have been carried out on the collapse and failure modes of underground caverns. Loucks et al. [9] theoretically believed that due to the influence of the upper load on the cave, tensile stress and tensile cracks are generated at the roof of the cave, and the sidewall of the cave is subjected to shear stress, resulting in shear cracks in the sidewall. As the load continues to increase, the crack propagation range continues to increase, which eventually leads to the collapse of the cave roof and cave. Zhu et al. [10] proposed a new model for the compaction failure of carbonate rock, explaining the mechanism of carbonate rock cavern failure from a microscopic perspective. Tang et al. [11] systematically summarized the structural characteristics of ancient karst collapse and believed that modern karst collapse was mainly due to the collapse of the roof and side walls caused by cracks under the action of gravity. At present, scholars at home and abroad have studied the collapse and failure of karst caverns mainly by analyzing and judging the collapse and failure phenomenon of karst caves through drilling cores and imaging logging data. The conditions and mechanical causes of collapse and failure of reservoirs are not very clear, and thus require further research. Simultaneously, a more mature theory has been proposed for the expansion and closure mechanism of fractures. However, the scientific issue of the interaction mechanism between caves and fractures in the process of reservoir development needs to be further studied. Considering the effects of reservoir depletion on wellbore stability, Aadnoy [12] established a model considering the increase in surrounding rock stress when the pore fluid was withdrawn and obtained the change law in critical wellbore collapse and fracturing pressure when the reservoir pore pressure decreased. Chen et al. [13] proposed six types of reservoir seepage models based on the combination of fractures and caves in fractured-cavity carbonate reservoirs in Tahe Oilfield. The research results indicate that different types of reservoirs show different mining characteristics. Chen [14] used this to predict the plane distribution of different reservoirs; Wang et al. [15] used the unit split method to simulate how caves in fracture-cavity carbonate reservoirs affect hydraulic crack extension and found that hydraulic cracks and cavern reservoirs can communicate effectively only when there is a large difference in the in situ stress. Yang [16] proposed a "fracture-cavity unit" through 3D seismic monitoring data, which more reasonably explained the dynamic characteristics of the mining process of carbonate reservoirs. Liu et al. [17] carried out a physical simulation experiment of hydraulic fracturing under a true triaxial stress state, using acoustic emission devices to monitor the hydraulic fracture propagation in real-time, and obtained the interaction mode between the cave and the hydraulic fracture and the key influence of the interaction factors. Compared with the actual situation, not only are the current numerical simulation studies mostly limited to one-way influences, but there is also a lack of large-scale geomechanical model tests to grasp the overall degeneration characteristics and failure laws of the caves and fractures, which leads the research of the interaction between the caves and fractures to be not thorough.

Based on previous studies, from the perspective of the numerical simulation and model tests, we used the finite-element software RFPA and a true three-dimensional geomechanical model test and carried out numerical calculations of collapse and failure of the karstic caves in fractured-cavity oil reservoirs. The model test results obtained the deformation characteristics and collapse failure mode of karst caves in unfilled fracture-cavity reservoirs and revealed the interaction between voids and fractures and provided a theoretical basis for oil extraction in Tahe fracture-cavity reservoirs.

## 2. Physical Model Test

### 2.1. Project Background

Tahe Oilfield is located in Kuche County and Luntai County, Xinjiang Uygur Autonomous Region, and is about 70 km northeast of Luntai County. The regional structure is located on the Aixike 2 structure in the southwest of the Akekule bulge in the middle

section of the secondary tectonic Shaya uplift in the Tarim Basin. The Potassic reservoirs are typical carbonate reservoirs buried at depths between 5300 m and 6200 m. Cracks and caves are very developed. Most of the caves in the reservoir are above 5 m and within 20 m, of which the majority are about 10 m [18–24]. They are extra-large, ultra-deep, low-abundance heavy oil reservoirs. In the process of oil recovery, with the decrease of the formation pressure, the collapse of karst caves or closure of large, fractured oil production channels occurs downhole, which severely affects oil well production and the recovery of oil reservoirs. On this basis, to avoid the impact of cave collapse and fracture closure on the production of oil wells, we carried out an experimental study on the cave stability of Ordovician carbonate fracture-cavity reservoirs in Tahe Oilfield.

### 2.2. The Principles of Similitude

The geomechanical model test is a method of scaling down geological engineering problems according to similitude theory. The geomechanical model is a reproduction of real physical entities, and it can genuinely reflect the spatial relationship between geological structure and engineering structure based on meeting similar theories. It can more accurately simulate the deformation and failure process of the surrounding rocks of underground caverns [25–27]. In this paper, a reduced-scale geomechanical model is used to simulate karst caves in fracture-cavity reservoirs, which meet the following similar conditions:

$$C_\sigma = C_r C_L \tag{1}$$

$$C_\delta = C_\varepsilon C_L \tag{2}$$

$$C_\sigma = C_\varepsilon C_E \tag{3}$$

$$C_\sigma = C_E = C_c \tag{4}$$

where $C_\sigma$, $C_r$, $C_L$, $C_\delta$, $C_\varepsilon$, $C_E$ are stress similarity scales, bulk similarity scales, geometric similarity scales, displacement similarity scales, strain similarity scales, elastic modulus similarity scales.

The geomechanical model test requires that the similarity scale of all dimensionless physical quantities (such as strain, internal friction angle, friction coefficient, Poisson's ratio, etc.) is equal to 1, and the similarity scale of physical quantities of the same dimension is equal, that is:

$$C_\varepsilon = 1, \ C_f = 1, \ C_\phi = 1, \ C_\mu = 1 \tag{5}$$

where $C_f$, $C_\phi$, $C_\mu$ are similar coefficient of friction, similar scale of friction angle, and similar scale of displacement.

### 2.3. Development of Similar Material

In general, the larger the geometric similarity scale of a model is, the higher is the accuracy, and the more it can reflect the actual situation of the prototype, but the cost of testing and time required will be higher. The determination of the geometric similarity scale will also be affected by external dimensions of the model, its stability, and the effects of loading equipment. In this model test, considering the engineering and the loading system's situation, 1:50 was selected as the similarity scale for this test. The physical model is set to a cube with a side length of 700 mm. Through laboratory tests, mechanical parameters such as compressive strength, tensile strength, elastic modulus, Poisson's ratio, cohesion, and internal friction angle of carbonate rocks in Tahe fracture-cavity reservoirs were accurately obtained (Figure 1). The specific parameters are shown in Table 1. Self-developed ferrite sand-cemented, geotechnically similar material [28] was used to configure the model, as similar materials were required for this model test to meet similar conditions (Table 2). By conducting many proportioning tests, ultimately, the component ratio of iron

powder, barite powder, and silica sand was determined to be 1:0.67:0.25. The cementing agent concentration was 17%, and the cementing agent as a percentage of aggregate weight was 6%.

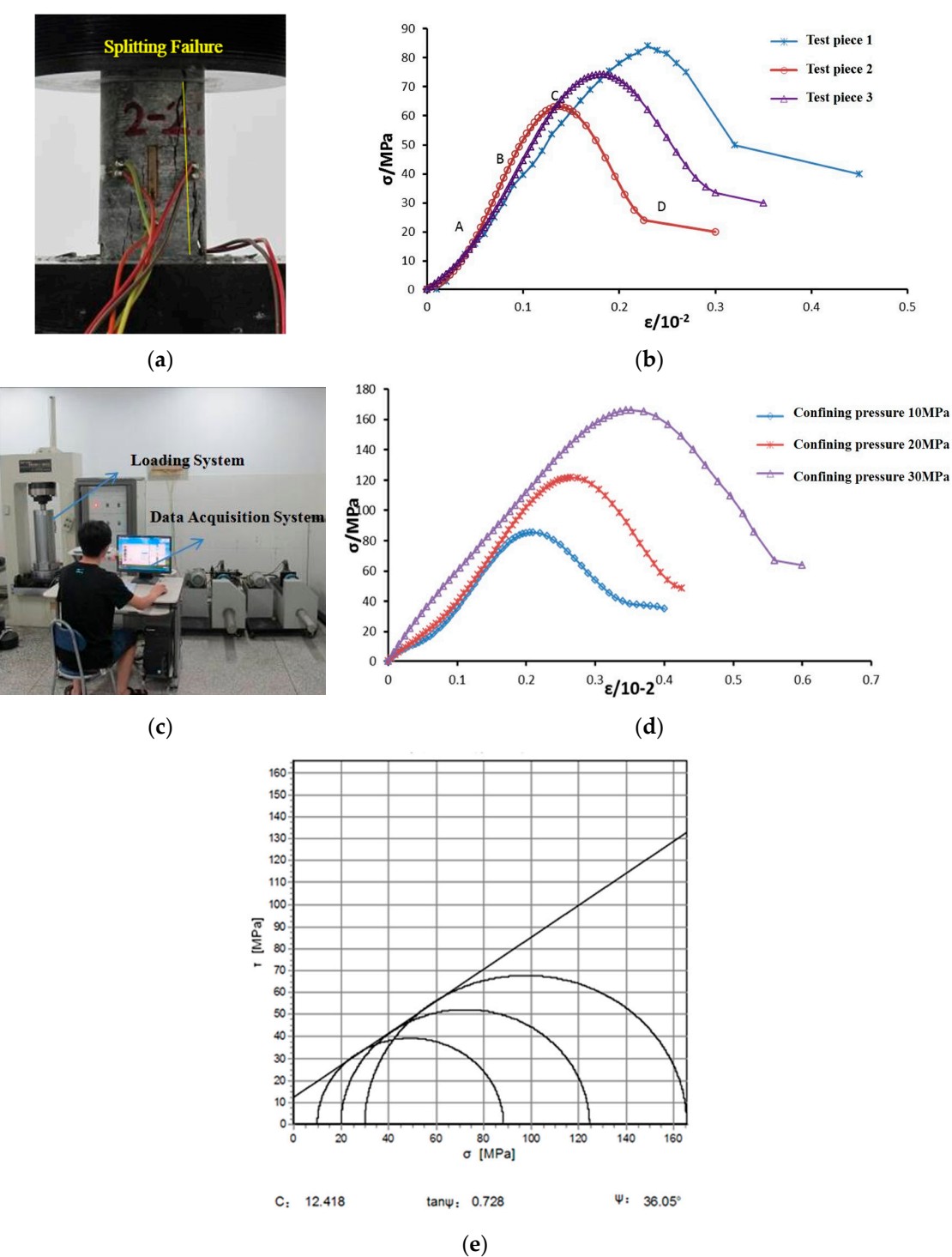

**Figure 1.** The results of the laboratory tests. (**a**) Uniaxial compression test; (**b**) uniaxial compression stress-strain curve; (**c**) triaxial compression test; (**d**) triaxial compression stress-strain curve (**e**) Mohr stress circle intensity curve.

**Table 1.** Physical and mechanical parameters of the original rock.

| Material | Bulkdensity (KN/m$^3$) | Elastic Modulus (GPa) | Compressive Strength (MPa) | Tensile Strength (MPa) | Cohesion (MPa) | Internal Friction Angle (°) | Poisson's Ratio |
|---|---|---|---|---|---|---|---|
| Carbonate | 27 | 36 | 74 | 3.8 | 12 | 36 | 0.25 |

**Table 2.** Theoretical calculations of physical and mechanical parameters of the model's similar materials.

| Similar Material | Bulkdensity (KN/m$^3$) | Elastic Modulus (GPa) | Compressive Strength (MPa) | Tensile Strength (MPa) | Cohesion (MPa) | Internal Friction Angle (°) | Poisson's Ratio |
|---|---|---|---|---|---|---|---|
| Carbonate | 26.8~27.1 | 710~820 | 1.38~1.61 | 0.71~0.82 | 0.23~0.26 | 35.4~36.5 | 0.23~0.26 |

*2.4. Model Test Plan*

In combination with the actual operating conditions of Tahe Oilfield, the self-developed true three-dimensional geomechanical model test system for high in situ stress is used to load the initial in situ stress in the cave area. This system mainly consists of the following parts: true 3D loading system and reaction frame, the intelligent hydraulic controlling system, and the automatic data acquisition system, as shown in Figure 2. The external dimensions of the reaction device of this system are 2.0 m in length, 1.75 m in height, and 1.75 m in width; the model size is 0.7 m in length, 0.7 m in height, and 0.7 m in width; the model geometric similarity scale is 50; and the simulation range of the design model test is: length × width × height = 35 m × 35 m × 35 m. The prototype cave size is: diameter = 5 m, model cave diameter = 100 mm. The vertically-arranged model high-angle crack size is length × height × thickness = 100 mm × 60 mm × 10 mm, and the model crack spacing is 60 mm. The ground stress in the prototype cave area is vertical stress $\sigma_1$ × maximum horizontal principal stress $\sigma_2$ × minimum horizontal principal stress $\sigma_3$ = 150 MPa × 90 MPa × 54 MPa, and the ground stress in the model cave area is vertical stress σ1 × maximum horizontal principal stress $\sigma_2$ × minimum horizontal principal stress $\sigma_3$ = 3 MPa × 1.8 MPa × 1.08 MPa. Figure 3 is a schematic diagram of loading a geological model (Figure 3 and Table 3).

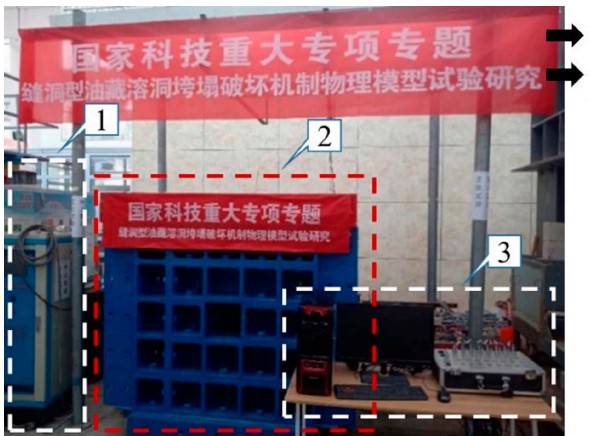

**Figure 2.** True 3D geomechanical model test system. 1: the intelligent hydraulic controlling system; 2: 3D loading system and reaction frame; 3: the automatic data acquisition system.

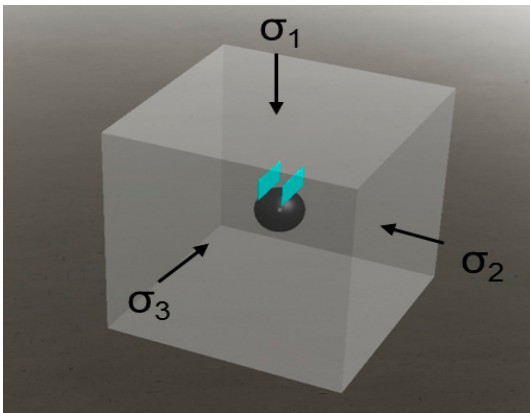

**Figure 3.** Geological model loading diagram.

**Table 3.** Loading steps corresponding to the loading stress.

| Loading Steps | 1 | 2 | 3 | 4 | 5 | 6 | 7 | 8 | 9 | 10 |
|---|---|---|---|---|---|---|---|---|---|---|
| $\sigma_1/\text{MPa}$ | 15 | 30 | 45 | 60 | 75 | 90 | 105 | 120 | 135 | 150 |
| $\sigma_2/\text{MPa}$ | 5.4 | 10.8 | 16.2 | 21.6 | 27 | 32.4 | 37.8 | 43.2 | 48.6 | 54 |
| $\sigma_3/\text{MPa}$ | 9 | 18 | 27 | 36 | 45 | 54 | 63 | 72 | 81 | 90 |

*2.5. Layout of the Monitoring Sensor*

In order to effectively observe the deformation changes during loading of the karst cave, monitoring points are arranged in the typical parts of the model body. The miniature multi-point displacement meter is used to monitor the deformation of the circumference of the hole and the crack. There are five measuring points embedded in each measuring point, and the distance from the cave wall is 5 mm, 50 mm, 100 mm, 200 mm, and 300 mm. The distance between the pressure cell measuring lines around the crack is 50 mm, and the distance between each point in the measuring line is 30 mm. The positions of the monitoring sensors can be found in Figures 4 and 5.

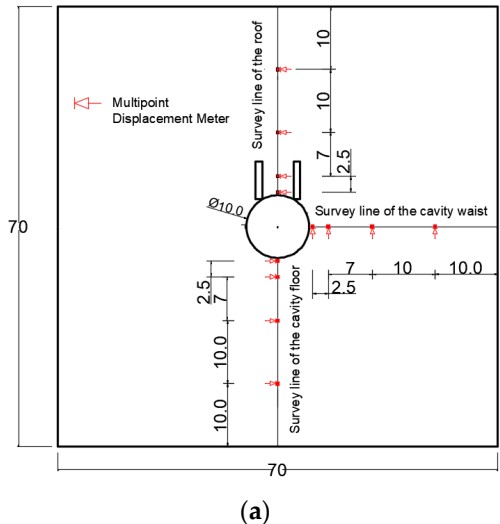

(**a**)

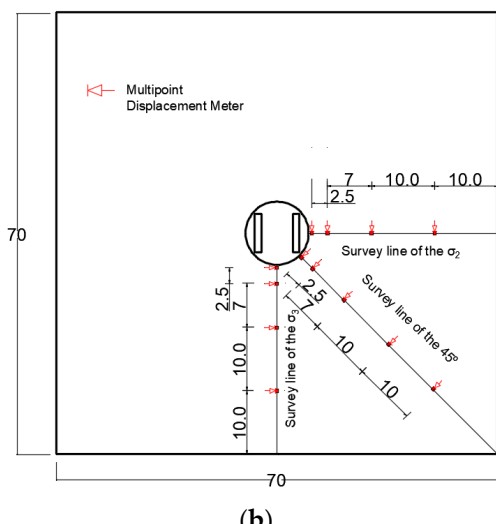

(**b**)

**Figure 4.** Schematic diagram of the specific position of the displacement measurement lines. (**a**) Front view; (**b**) top view.

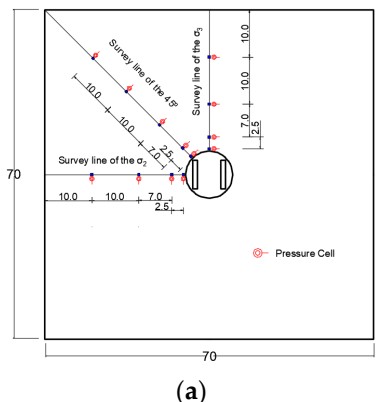

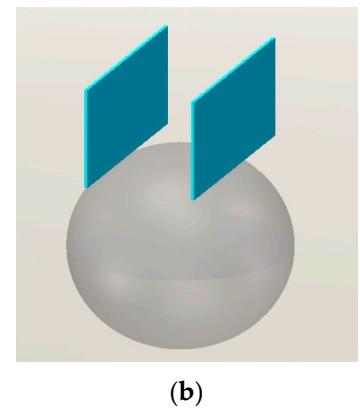

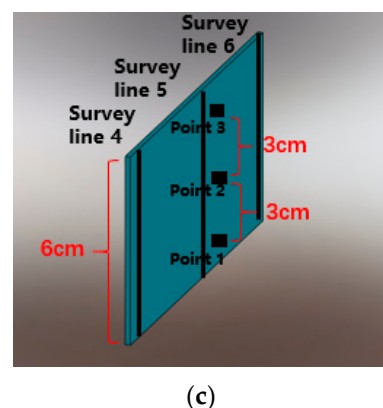

(**a**)  (**b**)  (**c**)

**Figure 5.** Schematic diagram of the specific location of the stress measurement lines. (**a**) Top view; (**b**) relative crack location; (**c**) layout of stress measurement points around cracks.

### 2.6. Construction of the Geomechanical Model

The test model is made by layered compaction technology [29]. The model is divided into ten layers, each with a thickness of 70 mm. The production process is as follows. First, the materials are mixed uniformly according to the designed ratio, and then evenly spread in the model test device after weighing. Then, the uniformly spread model materials are compacted according to the determined pressure, and an air blower is used to dry the compacted model material. When the height of the sensor is preset, the sensor is cut into a groove (Figure 6a); the above steps are repeated until the fifth layer is completed. After the current five-layer model body is completed, a pre-made ice ball containing a hollow metal tube with the same shape and size as the cave is completely wrapped with PVC film and buried in the center position, and similar materials are continued to be filled until the paraffin body is completely covered. A heating rod is inserted into the metal tube and pulled out after the ice has completely melted; then the PVC film is removed (Figure 6b). At this time, a cavity is formed. A PVC film three times the size of the hole is placed over the hole left by the metal tube and is pressed with similar materials, to continue completing the model body located on the upper part of the cave from the corners. When presetting the cracks, 2 PTFE sheets with a total thickness equal to the thickness of the crack and 2 and 1.5 times the height of the crack must be made in advance. The PTFE sheet is placed at the position of the crack. Vaseline is applied to the surrounding area in advance. After the material is compacted, the higher PTFE plate is gently shaken left and right and pull out after it is loosened. During this process, attention must be paid to pressing similar materials on both sides of the PTFE plate to prevent damage when pulling it out with great force. The short PTFE board is then pulled out. At this time, a crack cavity is formed (Figure 6c).

A three-dimensional loading system is used to apply boundary loads to the model. The specific loading steps are shown in Table 3. At the end of each loading step, loading ceases, and the experimental data are observed. When the sensor data are stable, loading resumes in the next step until the end of the loading process.

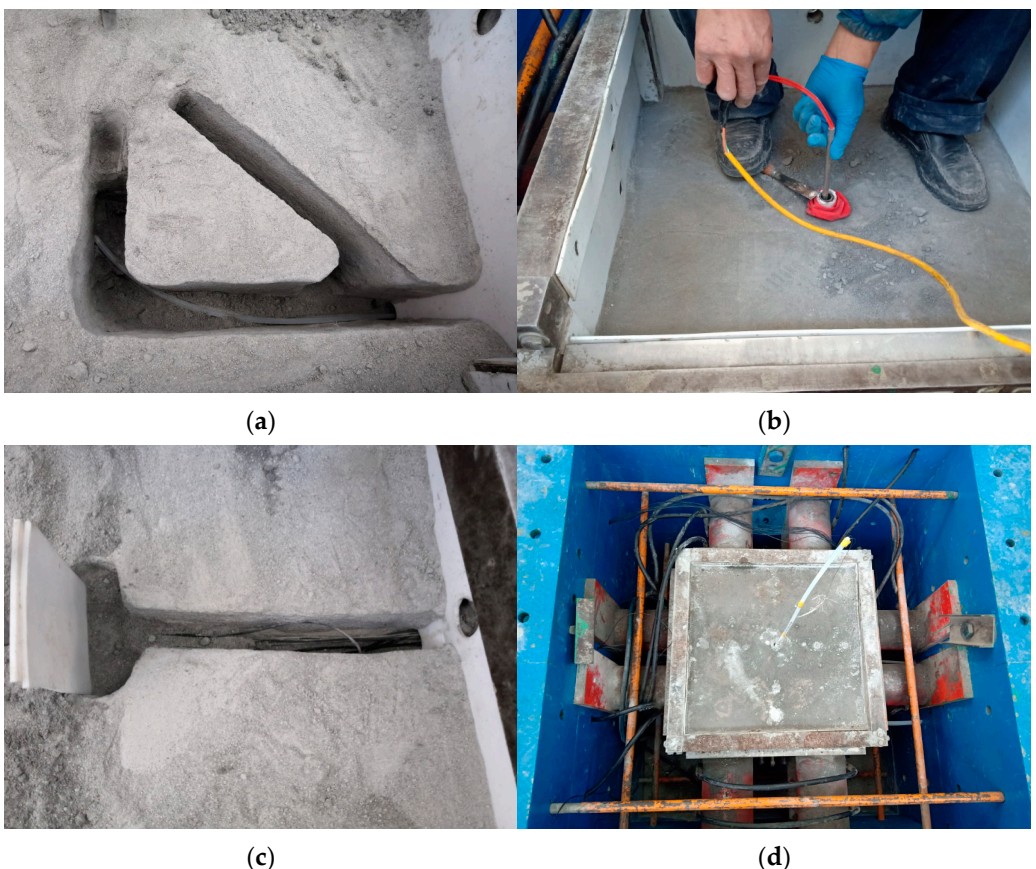

**Figure 6.** The procedure of model construction. (**a**) Embedding monitoring elements; (**b**) melting ice to form a cavity; (**c**) precasting cracks; (**d**) completing the model.

## 3. Model Test Results

### 3.1. Variation in the Displacement

From the analysis of Figures 7 and 8, under the influence of the middle and late Caledonian tectonic and deadweight stress, when the loading stress reaches 50% of the final loading stress, the displacement of the cave roof suddenly increases from 53 mm to 120 mm, and the displacement increases by 126%, indicating that cracks occur in the roof of the cave; when the final stress is loaded, the displacement curve abruptly changes, and the displacement of the cave roof increases by 600%, indicating that the roof of the karst cave falls, and the unfilled cave eventually collapses.

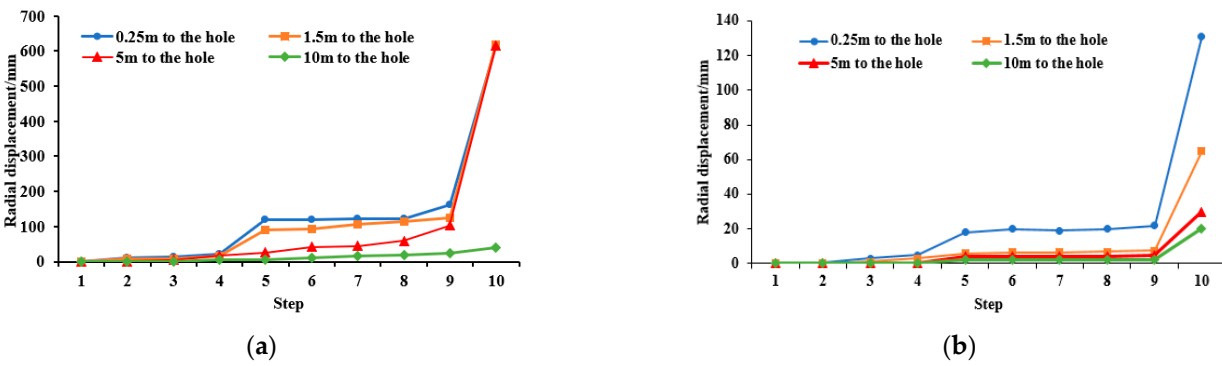

**Figure 7.** Curve of radial displacement of the measurement points around the cave accompanied by loading steps. (**a**) Radial displacement change at the top of the cave; (**b**) change of radial displacement at the bottom of the cave.

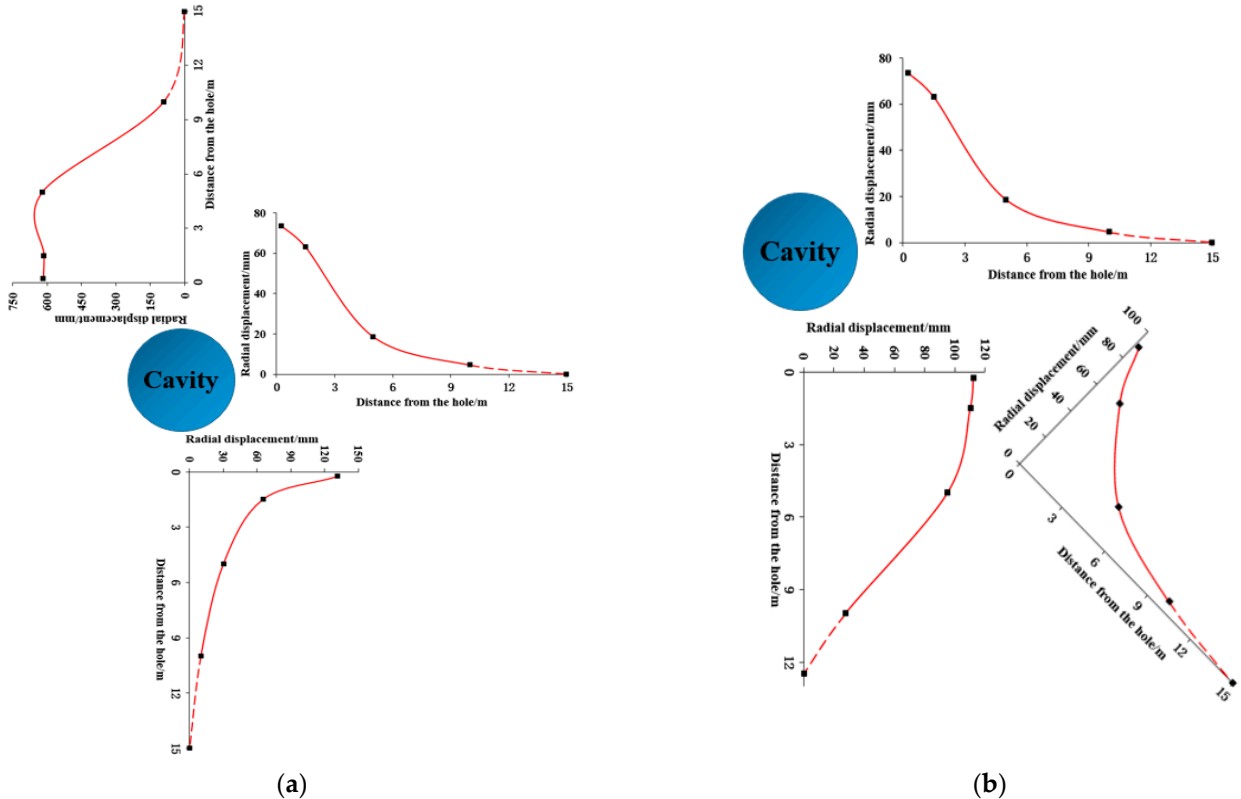

**Figure 8.** Distribution law of radial displacement around the cave. (**a**) Front view; (**b**) top view.

During the loading process, the cave shrinks in the circumferential direction of the cave. The closer the cave wall is, the larger is the radial displacement of the surrounding rock. The displacement of the surrounding rock within the range of 1 times the diameter of the cave changes significantly under the load's influence. The effect of stress loading on the deformation of the surrounding rock gradually decreases beyond this range. There is still a certain amount of displacement at the site of twice the cave's diameter, and finally the displacement at 3 times the diameter of the cave is approximately 0, which indicates that the impact range of the cave collapse is 2 to 3 times the hole diameter.

The deformation at the top of the cave is much larger than that at the other parts of the cave. The deformation is more uniform within 1.5 times the hole diameter, and it quickly decreases to 0 after 1.5 times the hole diameter. The result shows that the cave roof has partially collapsed, but the karst cave, especially the lower hemisphere, has not entirely collapsed and still has some stability.

### 3.2. Variation Features of the Stress

Figure 9 shows that during the loading process, the radial stress of the surrounding rock is released, and the closer the cave wall is, the smaller the radial stress is. As the distance from the cave wall becomes more extensive, the radial stress gradually increases to the original rock stress, and concentration of the surrounding rock's tangential stress occurs. As the distance from the cave wall increases from near to far, it increases first and then decreases, and finally tends to the original rock stress. A more extensive pressure arch is formed in the depth of the surrounding rock. Stress transfers from the surrounding rock to the deep part of the surrounding rock.

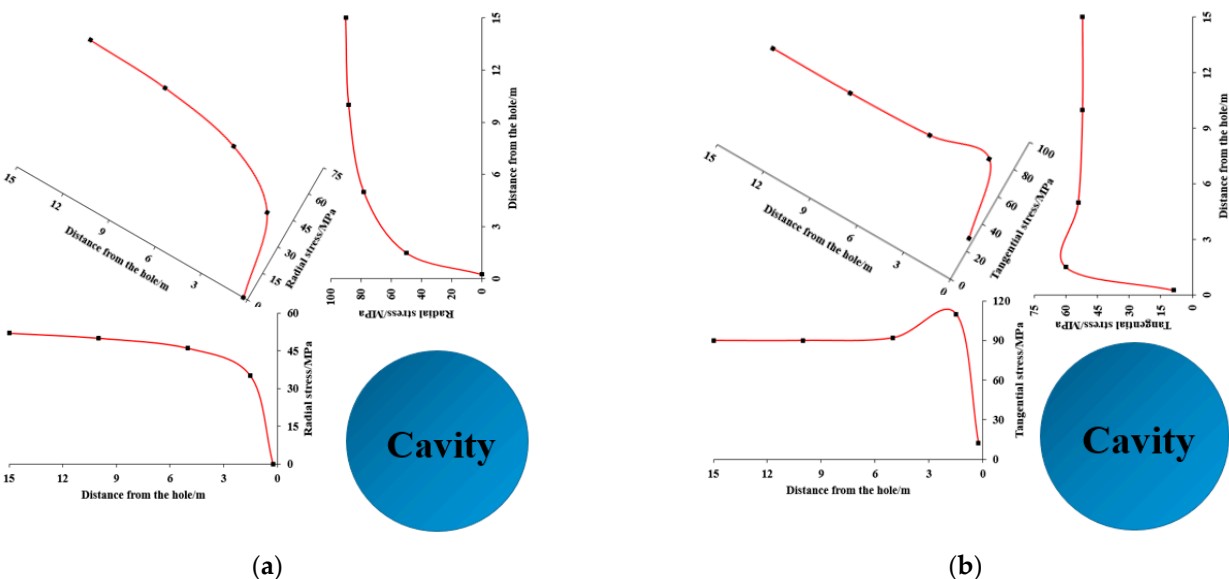

(**a**)　　　　　　　　　　　　　　　　　　(**b**)

**Figure 9.** Radial stress and tangential stress distribution around the maximum diameter hole. (**a**) Radial stress; (**b**) tangential stress.

During the loading process, the influence range of the loading stress around the cave is 2 to 3 times the hole diameter, and the stress of the surrounding rock changes sharply within 2 times the hole diameter. When the loading stress reaches 40% of the final, the tangential stress around the crack decreases sharply, indicating that cracks begin to close, and the crack stress releasing near the cave end is most apparent. The crack closure gradually extends from the lower end near the top of the cave to the upper end (Figure 10).

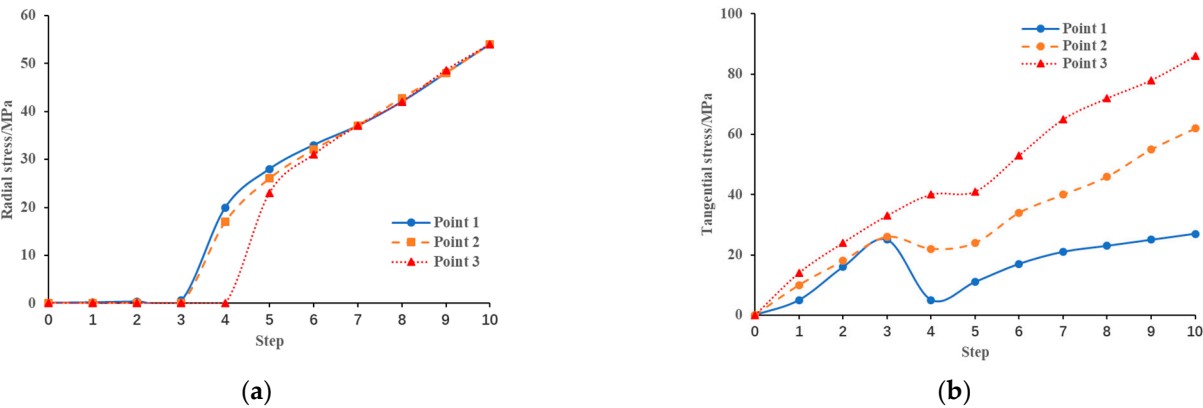

(**a**)　　　　　　　　　　　　　　　　　　(**b**)

**Figure 10.** Curve of stress on measuring point 4 around the crack with the loading step. (**a**) Radial stress; (**b**) tangential stress.

### 3.3. Failure Modes

In order to further verify the accuracy of the analysis results of the test data of the model test, the model body of the unfilled karst cave is opened and can be observed after the loading test. The cavity's failure situation is shown in Figure 11.

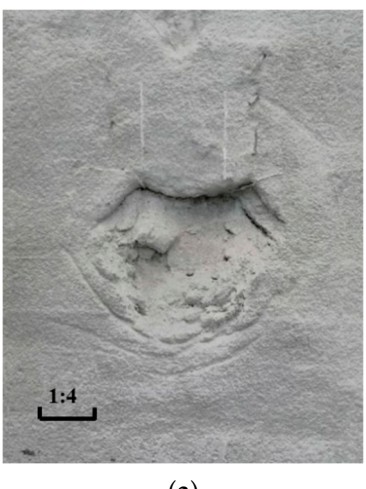
**(a)**

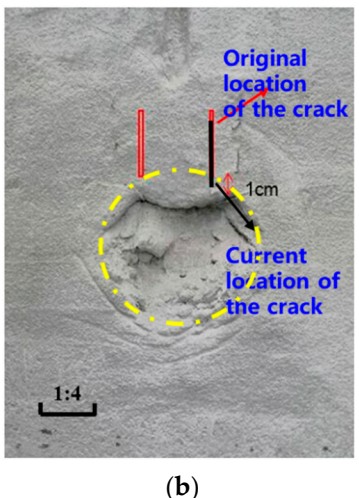
**(b)**

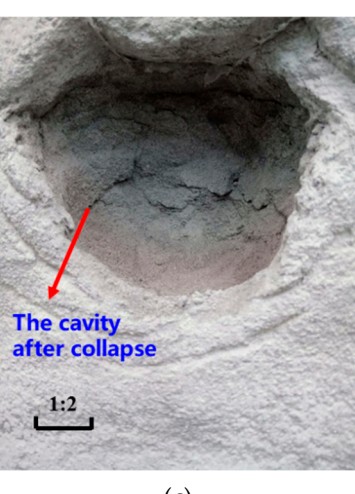
**(c)**

**Figure 11.** Fracture results after model test. (**a**) Model cutaway; (**b**) comparison of seam-hole joints before and after the test; (**c**) residual cavity after slag removal.

The roof of the karst cave is subjected to shear stress and compressive stress to produce shear cracks, and the existence of prefabricated cracks together weakens the bearing capacity of the roof, causing the roof to sink as a whole and collapse and damage some areas. The upper part of the karst cave collapses into an M shape, indicating that a slipping dislocation surface is generated when the roof collapses due to the existence of cracks, and it is also proven from the side that the prefabricated cracks reduced the strength of the roof. There is a shear slip failure line around the lower half of the cave. Two clusters of rupture zones intersect and cut each other around the bottom of the cave. A split failure zone surrounding the cave wall appears. After cleaning, it can be seen that it is still ellipsoidal, indicating the lower half compared with the upper half of the completely collapsed cave; considering that the weight of the roof is negligible relative to the loading stress, it proves that the prefabricated cracks reduce the bearing capacity of the roof.

*3.4. Conclusion of Model Test Research*

This test resulted in the following findings: ① As the distance from the cave wall increases, the deformation around the cave gradually decreases, the radial stress gradually increases and tends toward the original rock stress, and the tangential stress increases first and then decreases. Ultimately, it tends toward the original rock stress. ② The deformation at the top of the cave is much larger than that in the other parts of the cave, indicating that the collapse of the roof is the main manifestation of the collapse of the cave. The existence of prefabricated cracks weakens the bearing capacity of the roof, making the roof of the cave M-shaped after collapse. ③ Cracks occurred when the cave was loaded to 50% of the final loading stress, and collapse occurred when it was loaded to the final load. The prefabricated cracks were closed when the load reached 40% of the final load. The closure started from the near-hole end and gradually developed toward the upper end of the crack. ④ The displacement and stress changes show that the impact range of the collapse of unfilled caves with high-angle fractures is 2 to 3 times.

**4. Numerical Simulation**

*4.1. Setup of the Numerical Model*

This paper uses the finite element-based software RFPA to calculate and analyze the collapse of karst caves. This software combines mesomechanics and numerical methods to simulate rock deformation and nonlinear behavior of fractures by considering the heterogeneity of rock properties. It is a new type of numerical analysis tool for solving the problems of discontinuous medium mechanics by the continuous medium mechanics method. The discontinuous and irreversible behavior of material failure is simulated by

considering the weakening of the element's parameters (including stiffness degradation) after material failure. Therefore, the software can calculate and dynamically demonstrate the complete process of fractured-cavity reservoirs, from loading to rupture.

Figure 12 shows the numerical analysis model and boundary conditions of karst caves under two conditions. The additional stress caused by the overburden layer's gravity on the top of the karst cave was replaced by the vertical stress ($\sigma_3 = \gamma H$) multiplied by the average bulk density $\gamma$ and the actual buried depth H of the cave. Horizontal tectonic stress was applied to the left and right boundaries of the model. In this paper, H is 5500 m, $\sigma_3$ is 150 MPa, and horizontal structural stress is 90 MPa. The numerical calculation of the cave size is consistent with the actual cave size; that is, the model size is 35 m × 35 m, the cave diameter D is 5 m, the seam length h' is 3 m, the distance between the two cracks L is 2.5 m, and the bottom surface is a fixed boundary. Table 1 shows specific values of physical and mechanical parameters.

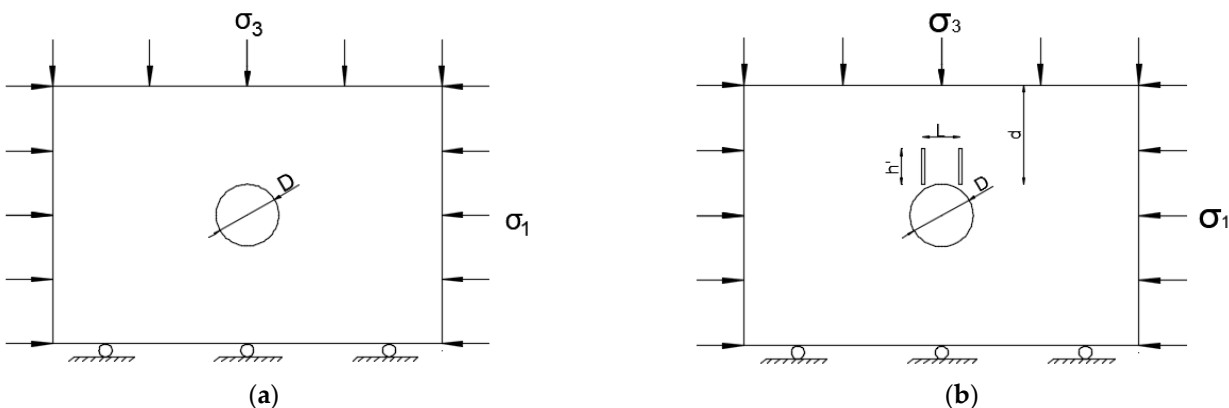

**Figure 12.** Numerical calculation models. (**a**) Cave without prefabricated cracks; (**b**) cave with prefabricated cracks.

### 4.2. Verification of Simulation Results

In order to verify the accuracy and matching of the results of the numerical simulations and the model test, we compared the model test and numerical simulation under the same stress condition. Comparisons are provided in Figure 13 regarding the variation in radial displacement and stress for the preset section. These figures show that the magnitude and trend changes of displacement and stress in the model test are basically consistent with the numerical simulation.

### 4.3. Analysis of Numerical Simulation Results

Figure 14 illustrates the failure mode of a circular karst cave under loading. The stress concentration at the waist is most apparent, and local damage gradually occurs. First, microcracks are generated. As the damage intensifies, the number of microcracks increases. Two clusters of oblique shear fracture zones are generated. These local failures reduce the roof's strength and bearing capacity. The roof gradually bends downward. The cave is slowly squashed and compacted during the deformation of the roof. The circular cave gradually becomes elliptical, and the fracture zone extends diagonally upward. Finally, the angle with the horizontal tectonic stress tends to be 45°, and eventually penetrates the cave's top. The roof plate no longer has bearing capacity, and the cave collapses. At this moment, the roof plate falls off as a whole and reaches equilibrium again after contacting the cave wall. There is still very little residual space in the cave. Simultaneously, the two clusters of fissure zones intersect at the bottom of the original cave to form a shear fracture zone at the bottom of the cave.

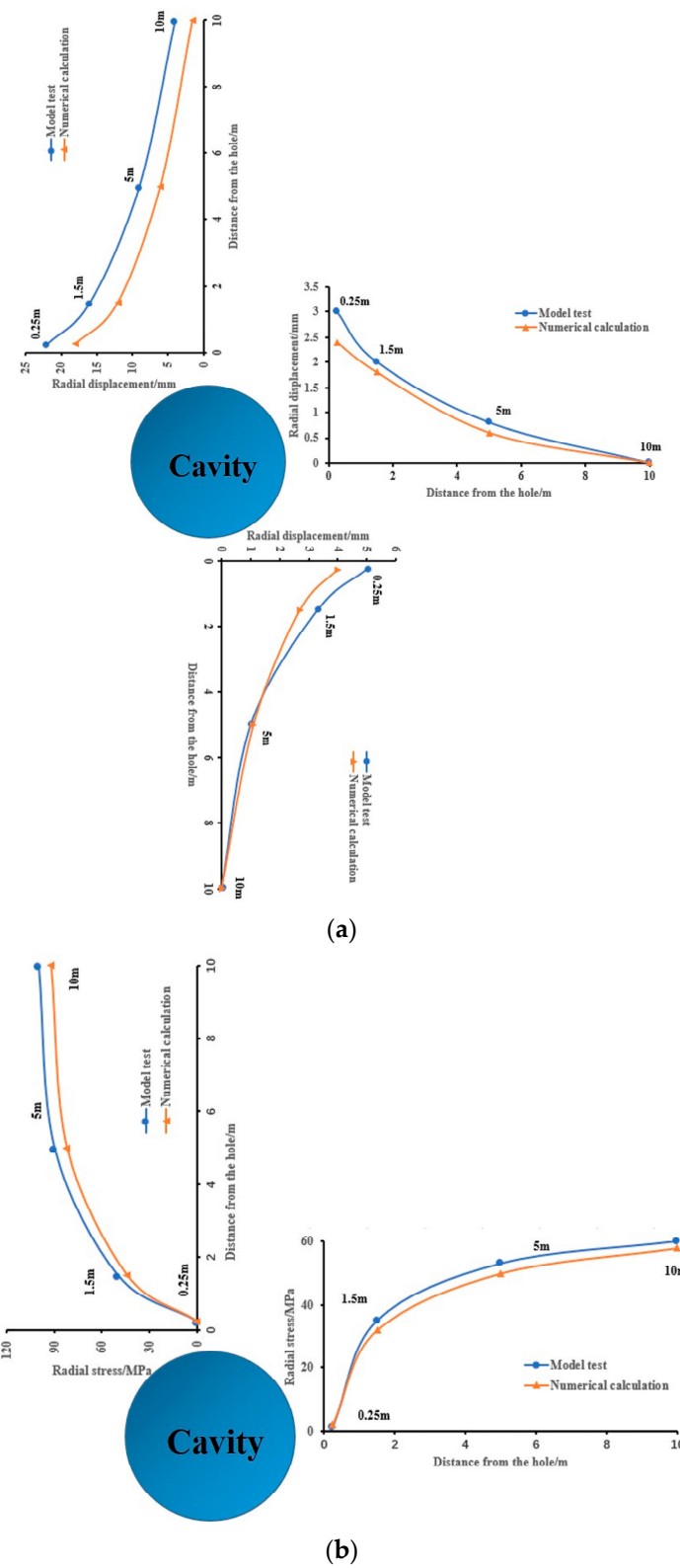

**Figure 13.** The comparison curves between the model test and the numerical simulations in terms of: (**a**) the displacement around the cavity; (**b**) the stress around the cavity and the crack.

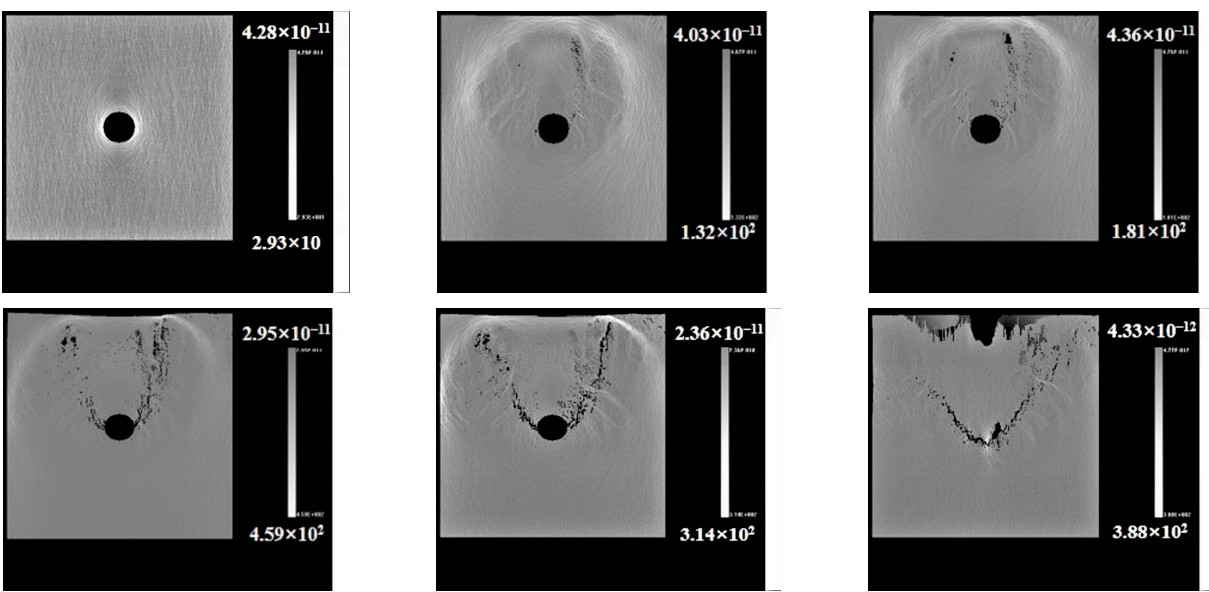

**Figure 14.** Collapse and failure process of karst cave without prefabricated cracks.

Figure 15 shows the collapse and failure process of the cave with prefabricated cracks. The crack tip first generates stress concentration. The wing-shaped tensile cracks develop on the upper tip of the prefabricated cracks, and the lower tip expands to the top of the cave. At the same time, local damage occurs at the waist of the cave. As the loading progresses, the wing-shaped tensile cracks and tensile cracks around the cave waist overlap and penetrate each other, forming a fracture zone that expanded obliquely upwards. Finally, the fracture zone penetrates the model boundary. At the same time, the lower tips of the prefabricated cracks extend to the cave, and the roof no longer has bearing capacity, so the cave collapses. The roof between the two cracks falls as a whole and reaches a state of equilibrium again after contacting the cave wall. The lower half of the cave does not wholly collapse but still has a certain strength. Therefore, the cave's residual space is large and is mainly distributed on both sides of the collapsed roof.

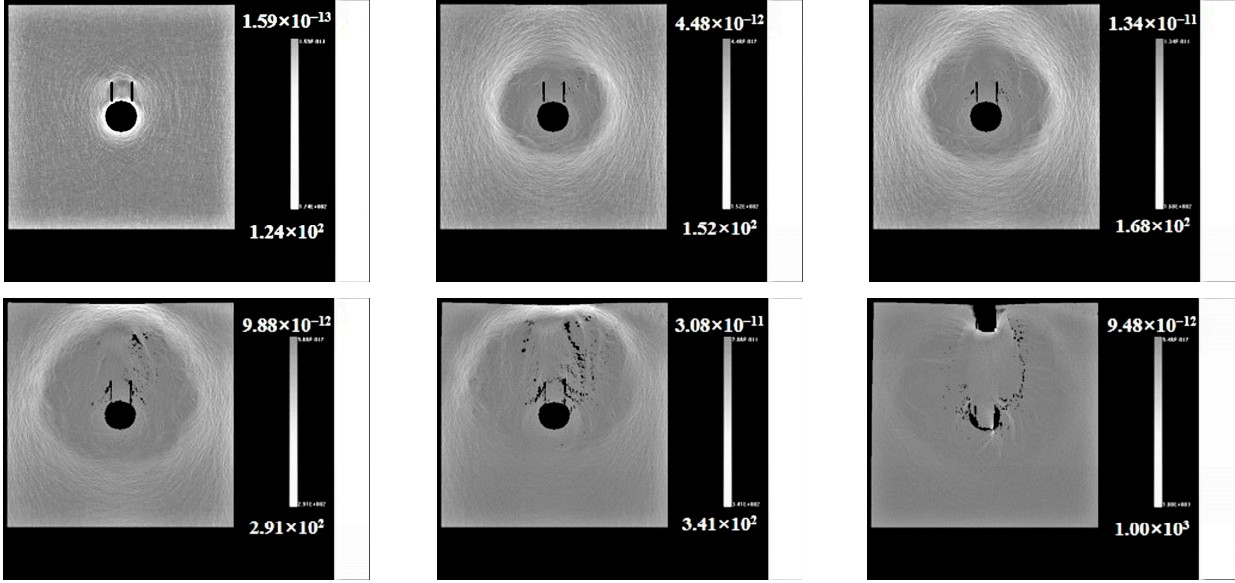

**Figure 15.** Collapse and failure process of the cave with prefabricated cracks.

It can be seen from the collapse and failure process diagram of the cave without prefabricated cracks that: ① The stress concentration area around the cave gradually expands outward, and the white area in the figure is the shear stress concentration area. ② The collapse of the cavity not only causes the roof to rupture and collapse but also has an impact on the stability of the surrounding rocks. The range of stress expansion when the cave collapses is the scope of the cave collapse. According to the collapse scope of the cave and the schematic diagram of the residual space in Figure 16, it can be seen that the scope of the collapse of the cave without prefabricated cracks has exceeded 3 times the diameter of the cave. The residual space after the cave collapse is not apparent, and the distribution is relatively scattered, which indicates that the cave's failure without prefabricated cracks is complete. The cave's collapse with prefabricated cracks affects 2.2 times the hole diameter, and there is a sizeable residual space on both sides of the collapsed roof, indicating that the roof of the cave located in the middle of the two cracks weakens its bearing capacity due to the existence of cracks. The karst cave partially collapses.

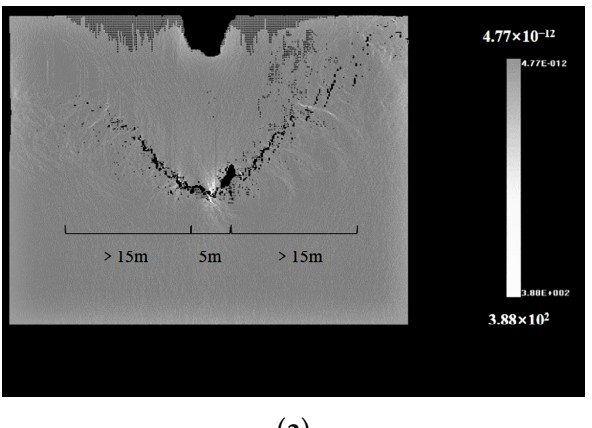

(**a**)

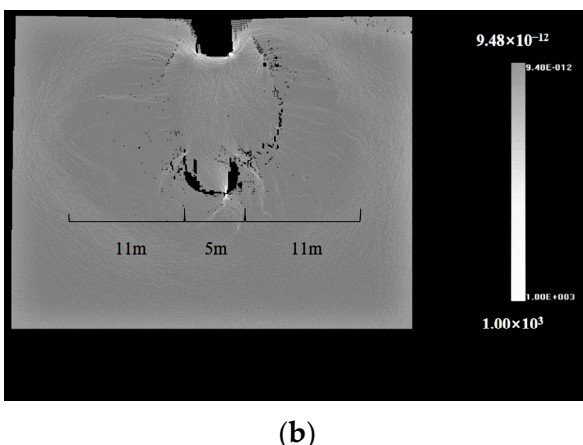

(**b**)

**Figure 16.** Impact range of cave collapse. (**a**) Cave without prefabricated cracks; (**b**) cave with prefabricated cracks.

By comparing the maximum principal stress cloud diagrams under the two conditions (Figure 17), it can be seen that the cave without prefabricated cracks generates enormous tensile stress at the waist of the cave. The tensile cracks gradually increase and expand diagonally upward, while the cave's tensile stress with prefabricated cracks is mainly concentrated at the crack's tip. The tensile cracks generated at the upper tip gradually expand and eventually intersect with the tensile cracks developed at the cave's waist. Simultaneously, the tensile stress concentration area is formed above the top of the cave at the middle of the two cracks. Compared to other parts, collapse and damage are more likely to occur, consistent with the analysis results above.

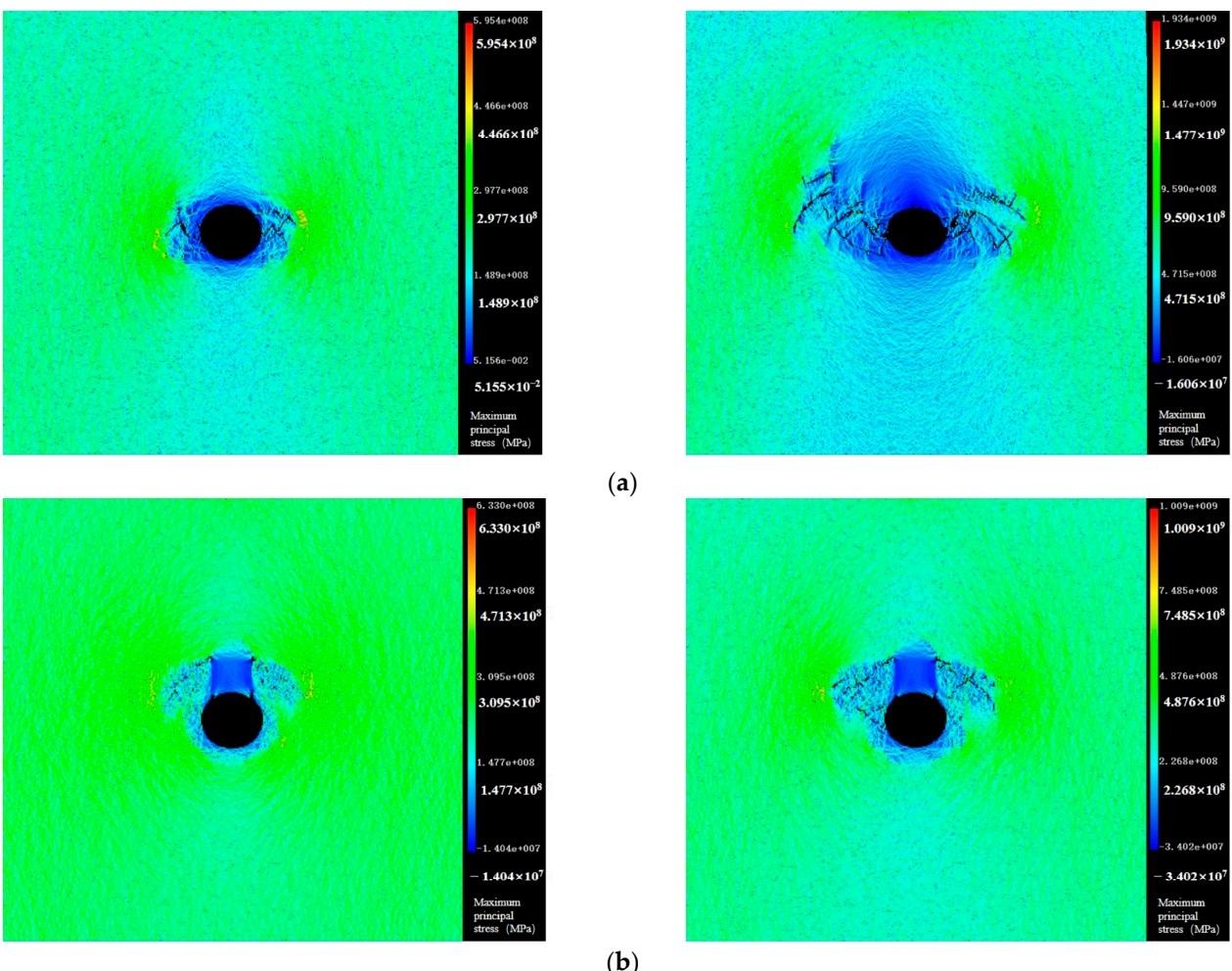

**Figure 17.** Cloud diagram of maximum principal stress during collapse and failure of karst cave. (**a**) Maximum principal stress cloud diagram of the cave's collapse process without prefabricated cracks; (**b**) maximum principal stress cloud diagram of the cave's collapse process with prefabricated cracks.

## 5. Discussion

Through the combination of a true three-dimensional model test and numerical simulation, we investigated the cause of the collapse of fractured-vuggy reservoirs and analyzed the interaction between fractures and caves. By analyzing the law of displacement change and comparing the impact range of collapse and opening the model body, we found the collapse failure mechanism of the cave, which proved that the prefabricated cracks promote the collapse of the cave, and the cave also has an impact on the closure law of the crack. Because the caves were too deep, previous scholars could only analyze the drilling core by pure numerical simulation and imaging logging data [11,21,23,24]. There is no discussion on the cause and mechanism of cave collapse. We used a true three-dimensional geomechanical model. The combination of experiment and numerical simulation makes up for the deficiencies of previous studies. However, due to the small number of experimental conditions, there has not been sufficient in-depth study on the partial cave collapse parameters and the filling state. Next, we will focus on the thickness of the roof, the size of the cave, and the angle and size of the prefabricated crack, as well as the filling medium of the cave and the filling degree. Further research will make the application of our research results more extensive and accurate.

## 6. Conclusions

Through the systematic research in this article, we can draw several important conclusions as follows:

(1) This paper uses the typical karstic caves in fractured-cavity oil reservoirs in Xinjiang Tahe Oilfield as the research background. The model test is used to study the collapse and failure mechanism of karst cave in the large and deep reservoir during reservoir exploitation. The test results effectively reveal the law of displacement and stress changes and the collapse mechanism of karst caves in fracture-cavity reservoirs.

(2) Model test results show that: ① As the distance from the cave wall increases, the surrounding rock deformation gradually decreases until there is no effect. ② After loading, the radial stress releases and tangential stress of the cave wall increases. As the cave wall's distance increases, the surrounding rock stress gradually tends to the original rock stress. ③ The vertical arrangement of the cracks first closes from the near end of the cave and progressively develops to close the cave's far end with the loading process.

(3) Using the software RFPA, the collapse failure modes and the cave's impact scope with and without prefabricated cracks are obtained. Moreover, the influence of the existence of prefabricated cracks on the collapse failure of karst caves is revealed. The calculation results show that prefabricated cracks reduce the karst cave's bearing capacity, making the collapse of the karst cave with prefabricated cracks incomplete, and the impact area of failure is much smaller than that of the cave without prefabricated cracks.

**Author Contributions:** Data curation, formal analysis, writing—original draft, writing—review and editing, Y.D.; methodology, project administration, writing—review and editing, Q.Z.; formal analysis, software, Writing—review and editing, W.X.; data curation and software, B.W.; methodology, writing—review and editing, X.L. and L.Z. All authors have read and agreed to the published version of the manuscript.

**Funding:** This study was financially supported by the National Science and Technology Major Project (NO: 33550000-18-ZC0611), the Natural Science Foundation of China (NO: 41772282), and the Taishan Scholars Project Foundation of Shandong Province and the National Key Research Development Project of China (No. 2016YFC0401804).

**Institutional Review Board Statement:** Not applicable.

**Informed Consent Statement:** Written informed consent has been obtained from the patient(s) to publish this paper.

**Data Availability Statement:** No new data were created.

**Acknowledgments:** This study was financially supported by the National Science and Technology Major Project (NO: 33550000-18-ZC0611T), the Natural Science Foundation of China (NO: 41772282), and the Taishan Scholars Project Foundation of Shandong Province and the National Key Research Development Project of China (No. 2016YFC0401804). We would also like to express our heartfelt thanks to those who contributed to the review of this article and to the editors.

**Conflicts of Interest:** The authors declare that they do not have any commercial or associative interest that represents a conflict of interest in connection with the work submitted.

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
