# Peer review of "Stability Analysis of Cavern Collapse in Fractured-Cavity Oil Reservoirs"

_sustainability, doi:10.3390/su15086809_

Round 1

Reviewer 1 Report

1. Figure 13 (b) is unclear and needs to be redrawn.
2. Please give further details on the materials and forming methods used in the production of the cavity of the model body.
3. The article has a compact structure. The physical test and numerical simulation verify each other. It has carried out an in-depth discussion on the collapse mechanism of the karst cave in the fracture-cavity reservoir, which has certain reference significance for the actual project. It is recommended to employ.

Reviewer 2 Report

The article is very interesting and well prepared. The article clearly presents the research problem, research methodology and the results obtained. The research results are interesting and usable by a wide spectrum of scientists. However, I am missing one important piece of information:

I admire the authors' desire to recreate the conditions of the collapse of the caverns as faithfully as possible. Unfortunately, you focus on caverns, meanwhile they are formed in rocks about which little is known. In the chapter "Project background" there is only a laconic phrase "The Potassic reservoirs are typical carbonate reservoirs..." and nothing else is known about these rocks. What does "typical" carbonates mean? - look at the classification of carbonates, e.g. Dunham - this is a whole group of rocks that differ in structure, texture and often lithology (eg dolomites and limestones). The lithology, structure and texture of rocks significantly affect the mechanical properties of rocks. Information whether the formation of these rocks is so constant that they can be considered homogeneous and isotropic should necessarily be included in the text! The prefabricated elements prepared for testing are approximately isotropic - is this also the case in reality? If they are mineralogically uniform rocks without stratification, are they porous? If they are porous limestones, the pore distribution affects the mechanical properties of the rock. Therefore, this information should also appear at the beginning of the article along with the lithological, structural and textural description of the limestones. If you use a simplification of the model in relation to the actual type of rocks to indicate the impact of the presence of cracks in the roof of caverns, write about it. Otherwise, the reader has the impression that the caverns occur in massive, homogeneous limestones of unknown characteristics.

In the Discussion chapter - in the conclusion you write: "Next, we will focus on the thickness of the roof / the size of the cave / the angle and size of the prefabricated crack, as well as the filling medium of the cave and the filling degree. Further research will make the application of our research results more extensive and accurate." I agree that the first step has been presented in this article, and I am glad that you intend to continue the research. However, further stages may start with the diversity of the rock environment, as it probably is in reality.

I also have some reservations about the drawings: Figure 11 - a graphic and numerical scale should be placed on each photo Figures 14, 15, 16, 17 - scale explanations are illegible

Reviewer 3 Report

1.      In this paper, the research on the mechanism of cave collapse and the law of reservoir fracture closure in fractured and vuggy reservoirs had been carried out by using the true three-dimensional geomechanical model test and numerical simulation software RFPA. The displacement, stress changes, and cave collapse failure mechanism in the process of cave collapse had been obtained, The collapse failure modes of seamless and the interaction mechanism between caves had been revealed. The manuscript was clearly structured and the language is generally understandable. The authors are advised to pay more attention to the language of this manuscript.

2.      To the reviewer's knowledge, similar geomechanical model tests have been conducted by the authors and by the same institute. Therefore, it is the authors' responsibility to highlight the novelty of this manuscript and the difference between this study and other tests conducted previously.

3.      Figure 13 (b) is unclear and needs to be redrawn.

4.      In Section 3, the authors conducted a set of numerical simulations and compared their test results with the modeling results, confirming that the simulation results are consistent with the test results. It is rather contradictory to me. Since the numerical simulation can reproduce the data measured from physical model test, what is the significance of the model test? Why not do the numerical simulations, which are more time- and cost-effective?

5.      As presented in the text, the physical model was constructed layer by layer. Will this construction process produce artificial interlayers within the model?
